# Manipulating midbrain dopamine neurons and reward-related behaviors with light-controllable nicotinic acetylcholine receptors

Romain Durand-de Cuttoli[1†], Sarah Mondoloni[1†], Fabio Marti[1], Damien Lemoine[1], Claire Nguyen[1], Jérémie Naudé[1], Thibaut d'Izarny-Gargas[1], Stéphanie Pons[2], Uwe Maskos[2], Dirk Trauner[3], Richard H Kramer[4], Philippe Faure[1‡*], Alexandre Mourot[1‡*]

[1]Neuroscience Paris Seine – Institut de Biologie Paris Seine (NPS – IBPS), Sorbonne Université, INSERM, CNRS, Paris, France; [2]Unité de Neurobiologie Intégrative des Systèmes Cholinergiques, Department of Neuroscience, Institut Pasteur, Paris, France; [3]Department of Chemistry, New York University, New York, United States; [4]Department of Molecular and Cell Biology, University of California Berkeley, Berkeley, United States

*For correspondence:
phfaure@gmail.com (PF);
almourot@gmail.com (AM)

†These authors contributed
equally to this work
‡These authors also contributed
equally to this work

Competing interests: The
authors declare that no
competing interests exist.

Reviewing editor: Olivier
Jacques Manzoni, Aix Marseille
Univ, INSERM, INMED, France

**Abstract** Dopamine (DA) neurons of the ventral tegmental area (VTA) integrate cholinergic inputs to regulate key functions such as motivation and goal-directed behaviors. Yet the temporal dynamic range and mechanism of action of acetylcholine (ACh) on the modulation of VTA circuits and reward-related behaviors are not known. Here, we used a chemical-genetic approach for rapid and precise optical manipulation of nicotinic neurotransmission in VTA neurons in living mice. We provide direct evidence that the ACh tone fine-tunes the firing properties of VTA DA neurons through β2-containing (β2*) nicotinic ACh receptors (nAChRs). Furthermore, locally photo-antagonizing these receptors in the VTA was sufficient to reversibly switch nicotine reinforcement on and off. By enabling control of nicotinic transmission in targeted brain circuits, this technology will help unravel the various physiological functions of nAChRs and may assist in the design of novel therapies relevant to neuropsychiatric disorders.
DOI: https://doi.org/10.7554/eLife.37487.001

## Introduction

Cholinergic neurotransmission provides a widespread and diffuse signal in the brain (*Picciotto et al., 2012*; *Sarter et al., 2009*). ACh alters neurotransmitter release from presynaptic terminals and affects neuronal integration and network activity, by acting through two classes of membrane receptors: metabotropic muscarinic receptors and ionotropic nicotinic ACh receptors (nAChRs). nAChRs consist of hetero- and homo-pentameric arrangements of α and β subunits (9 and 3 genes, respectively), yielding a high combinatorial diversity of channel composition, localization and function (*Zoli et al., 2015*). Nicotinic neuromodulation controls learning, memory and attention, and has been associated with the development of numerous neurological and psychiatric disorders, including epilepsy, schizophrenia, anxiety and nicotine addiction (*Taly et al., 2009*). Understanding how nAChRs mediate such diverse functions requires tools for controlling nicotinic neurotransmission in defined brain circuits.

ACh is a modulator of the VTA, a midbrain DAergic nucleus key in the processing of reward-related stimuli and in addiction (*Di Chiara and Imperato, 1988*; *Pignatelli and Bonci, 2015*;

**eLife digest** Acetylcholine is one of the most abundant chemicals in the brain, with key roles in learning, memory and attention. Neurons throughout the brain use acetylcholine to exchange messages. Acetylcholine binds to two different classes of receptors on neurons: nicotinic and muscarinic. As the name suggests, nicotinic receptors also respond to nicotine, the main addictive substance in tobacco, while muscarinic receptors respond to muscarine, present in certain poisonous mushrooms.

Nicotinic and muscarinic receptors each consist of many different subtypes. But standard pharmacology techniques cannot discriminate between the effects of acetylcholine binding to these different subtypes. Likewise, they cannot distinguish between acetylcholine binding to the same receptor subtype on different neurons. Durand-de Cuttoli, Mondoloni et al. have now developed a new nanotechnology that uses light to target specific acetylcholine receptor subtypes in freely moving mice.

The technology was tested in a brain region called the VTA, which is part of the brain's reward system. Experiments showed that when acetylcholine binds to a specific subtype of nicotinic receptors on VTA neurons – called β2-containing receptors – it makes the neurons release the brain's reward signal, dopamine. Switching these receptors on and off changed how the mice responded to nicotine. With the receptors switched on, mice preferred locations associated with nicotine. Switching the receptors off removed this preference. Nicotine may thus be addictive in part because it triggers VTA neurons to release dopamine via its actions on β2-containing nicotinic receptors.

This new technology will help reveal the mechanisms of action of acetylcholine and nicotine. Blocking the effects of nicotine at a specific time and place in the mouse brain may uncover the receptors and brain regions that drive nicotine consumption. Smoking remains a major cause of preventable death worldwide. This new approach could help us develop strategies to prevent or treat addiction.

DOI: https://doi.org/10.7554/eLife.37487.002

---

*Volkow and Morales, 2015*). The pedunculopontine and laterodorsal tegmental nuclei (PPN and LDT) are the two major cholinergic inputs to the VTA (*Beier et al., 2015*). Optogenetic activation of PPN and LDT neurons modulates the firing patterns of VTA DA cells and reward-associated behaviors (*Lammel et al., 2012*; *Dautan et al., 2016*; *Xiao et al., 2016*), implicating ACh in these processes. Yet, whether ACh directly affects neuronal excitability at the post-synaptic level, or whether it potentiates the release of other neurotransmitters through pre-synaptic nicotinic and muscarinic receptors is not known.

Brain nAChRs are expressed in high densities in the VTA, and in strategic places such as somatic and dendritic sites on GABAergic, glutamatergic and DAergic VTA cells, as well as on pre-synaptic terminals from extra-VTA afferents and from intra-VTA GABAergic interneurons (*Changeux, 2010*; *Zoli et al., 2015*). They are also present on DAergic terminals in the Nucleus Accumbens (NAc) and the prefrontal cortex (*Grady et al., 2007*; *Changeux, 2010*). Genetic and pharmacological manipulations have implicated VTA nAChRs in tuning the activity of DA neurons and in mediating the addictive properties of nicotine (*Mameli-Engvall et al., 2006*; *Maskos et al., 2005*; *Morel et al., 2014*; *Naudé et al., 2016*; *Picciotto et al., 1998*; *Tapper et al., 2004*; *Tolu et al., 2013*). However, understanding the mechanism by which ACh and nicotine participate in these activities requires to comprehend the spatio-temporal dynamics of nAChRs activation. Genetic manipulations can eliminate specific nAChRs, but they cannot provide kinetic information about the time course of nAChR signals that could be crucial for actuating VTA circuits and goal-oriented behaviors. Moreover, gene knockout can have unintended consequences, which include compensatory changes in expression of other receptors or channels, homeostatic adaptations and developmental impairments (*King et al., 2003*). Pharmacological agents allow activation or inhibition of nAChRs, but they diffuse slowly in vivo, they have limited subtype specificity and they cannot be targeted to genetically-defined neuronal cell types.

To fill this gap between molecular and circuit knowledge, we have developed the optogenetic pharmacology for rapid and reversible photocontrol of genetically-targeted mammalian neurotransmitter receptors (*Kramer et al., 2013*). We previously demonstrated light-controllable nAChRs (LinAChRs) in Xenopus occcytes, a heterologous expression system (*Tochitsky et al., 2012*). Here, we deployed strategies for acutely and reversibly controlling nicotinic transmission in the VTA in the mammalian brain, in vivo. β2* receptors account for the great majority of VTA nAChRs and are crucial for the pathophysiology of nicotine addiction (*Maskos et al., 2005*; *Faure et al., 2014*). We demonstrate acute interruption of nicotinic signaling in the VTA and reveal that endogenous pontine ACh strongly impacts on the firing patterns of VTA DA neurons. Moreover, we reversibly prevented the induction of nicotine preference in behaving mice by locally photo-antagonizing the effect of nicotine on VTA β2* nAChRs. This approach to optically antagonize neurotransmitter receptors in vivo will help sense the different temporal dynamics of ACh concentrations, and unravel the contribution of specific nAChR isoforms to nicotinic neuromodulation of neural circuits and associated behaviors, including drug abuse.

## Results

### Design and characterization of β2LinAChR

The vast majority of nAChRs in the mouse VTA contains the β2 subunit (*Zoli et al., 2015*; *Faure et al., 2014*). Therefore, we engineered this subunit to enable installation of light sensitivity. We transposed the rat β2E61C mutation, used previously in nAChRs expressed in Xenopus oocytes (*Tochitsky et al., 2012*), to the mouse β2 subunit to generate a photosensitizable receptor that traffics and functions normally in the mouse brain. The single cysteine-substitution, which is used for the anchoring of the photoswitchable tethered ligand Maleimide-Azobenzene-Homocholine (MAHoCh), faces the agonist binding sites (*Figure 1A*). MAHoCh has a photo-isomerizable azobenzene group, flanked on one side with a thiol-reactive maleimide moiety for conjugation to the cysteine, and on the other with a homocholine ligand for competitive antagonism of nAChRs (*Figure 1B*). In darkness, the azobenzene group adopts the thermally stable, extended *trans* configuration. Illumination with near-UV (e.g. 380 nm) light isomerizes the azobenzene core to the twisted, *cis* configuration. The *cis* isomer reverses to *trans* either slowly in darkness or rapidly in green light (e.g. 500 nm). Receptor activation in response to ACh agonist remained unaltered in darkness after conjugation of MAHoCh to β2E61C. However, agonist activation is blocked in 380 nm light, when *cis* MAHoCh occupies the agonist binding pocket (*Figure 1C*). Photo-control is bi-directional, and antagonism is relieved under 500 nm light when MAHoCh is in its *trans* form.

To verify whether nAChR currents could be photo-controlled, the β2E61C mutant was co-expressed with the WT α4 subunit in Neuro-2a cells (*Figure 1D*). Cells were treated with MAHoCh and any remaining untethered photoswitch was washed away prior to electrophysiological recordings. As expected, currents evoked by both carbamylcholine (CCh) and nicotine were strongly inhibited under 380 nm light, when tethered *cis* MAHoCh competes with the agonist (*Figure 1E*). Currents rapidly (<500 ms) and fully returned to their initial amplitude upon 525 nm light illumination. Repeated light flashes reduced and increased current amplitude without decrement, consistent with photochemical studies showing that azobenzenes are very resistant to photobleaching (*Szymański et al., 2013*). Spectroscopic measurements show that *cis* MAHoCh reverts to *trans* in darkness, but very slowly, with a half-life of 74 min in solution (*Tochitsky et al., 2012*). Consistent with this, we found that nAChR responses remained suppressed in darkness for at least ten minutes after a single flash of 380 nm light, but quickly recovered upon illumination with 525 nm light (*Figure 1F*). Hence, LinAChR could be rapidly toggled between its functional and antagonized forms upon brief illumination with the proper wavelength of light, but could also remain suppressed several minutes in darkness, eliminating the need for constant illumination.

### β2LinAChR enables inhibition of nicotinic currents in VTA DA neurons

We then tested whether nAChR currents could be photo-controlled in VTA DA neurons using β2LinAChR. To this aim, we virally targeted the cysteine-mutant β2 subunit together with eGFP under the control of the ubiquitous pGK promoter to the VTA of WT mice (*Figure 2A*). As expected, transgene expression was found at the injection site throughout the VTA both in TH+ and TH-

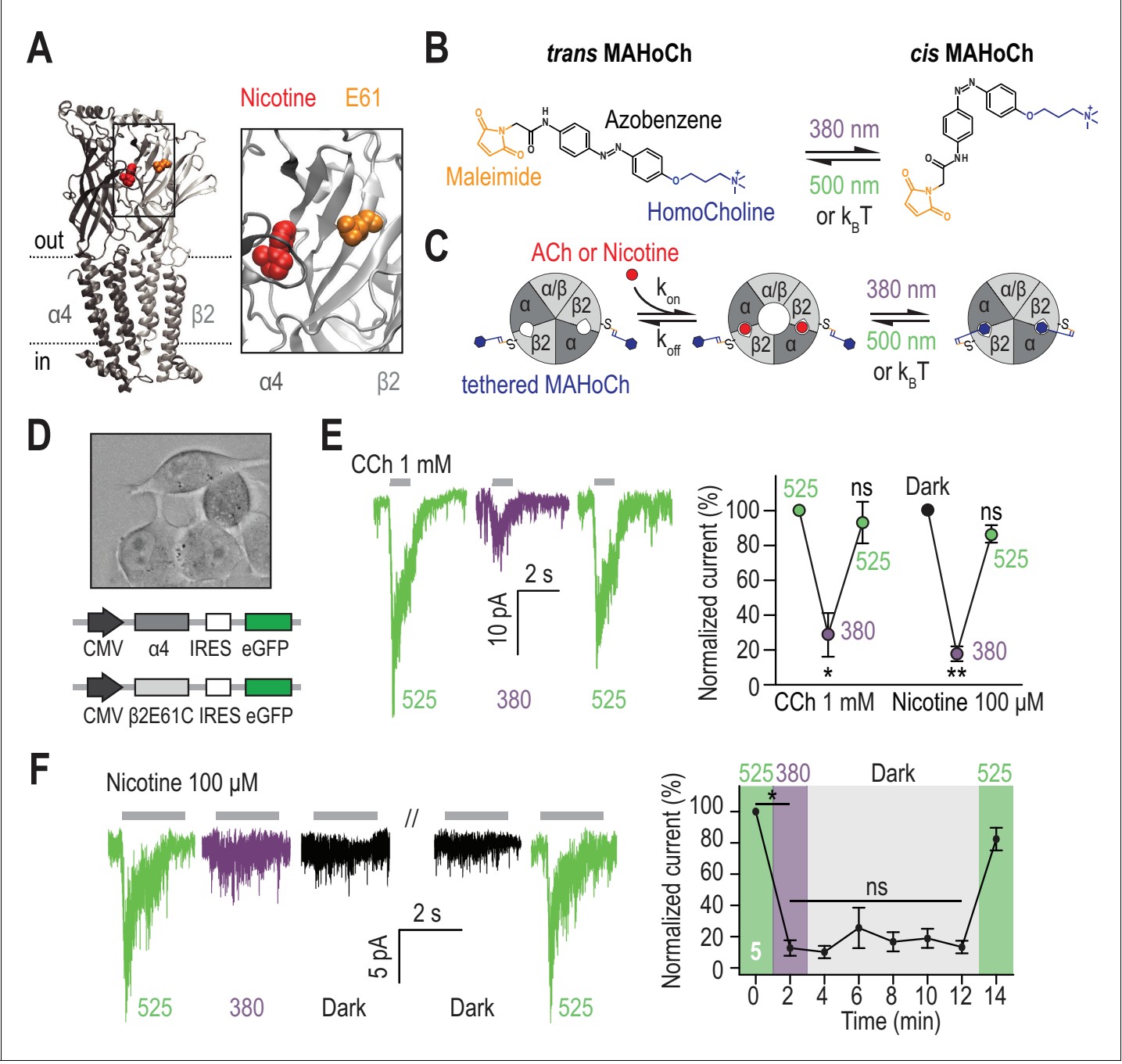

**Figure 1.** Design and characterization of β2LinAChR. (**A**) Crystal structure of the α4β2 nAChR (PDB ID 5KXI) (*Morales-Perez et al., 2016*) viewed parallel from the plasma membrane. The α4 subunit is in dark grey and the β2 subunit in light grey. The agonist binding sites are located in the extracellular binding domain, at the interface between the α and β subunits. Nicotine (red) and the amino acid E61 (orange) which has been mutated to cysteine in the β2LinAChR are represented as spheres. For clarity, only one αβ dimer is shown, and an extended view is shown on the right. (**B**) Chemical structure of *trans* and *cis* MAHoCh. The thiol-reactive group maleimide is shown in orange, the azobenzene photo-sensitive moiety in black, and the competitive antagonist homocholine in blue. In darkness, the azobenzene group adopts the thermally stable, extended *trans* configuration. Illumination with near-UV (380 nm) light photo-isomerizes the azobenzene core to the twisted, *cis* configuration. The *cis* isomer reverses to *trans* either slowly in dark conditions ($k_BT$) or rapidly under green light (500 nm). C*is-trans* photo-isomerization hence results in drastic changes in the geometry and end-to-end distance of MAHoCh. (**C**) Cartoon representation of β2LinAChR. MAHoCh is tethered to β2E61C, and the receptor still functions in the dark. Isomerizing the photoswitch back and forth between its *cis* and *trans* forms with two different wavelengths of light enables reversible photocontrol of the receptor: activatable under green light and antagonized under purple light. (**D**) Heterologous co-expression of α4 and β2E61C nAChR subunits in Neuro-2a cells. (**E**) Reversible photocontrol of α4β2LinAChR in Neuro-2a cells. Currents were recorded in whole-cell voltage-clamp mode at a potential

*Figure 1 continued on next page*

Figure 1 continued

of −60 mV and elicited by an application of CCh (1 mM, 1 s, n = 4) or nicotine (100 μM, 2 s, n = 5). Currents were strongly inhibited under 380 nm light (71.3 ± 12.5%, p=0.038 for CCh and 82.1 ± 4.2%, p=0.0082 for nicotine) and fully restored under 525 nm light (p=0.285 for CCh and 0.125 for nicotine). (F) Thermal stability of LinAChR photo-inhibition. After inhibition with 380 nm light, the amplitude of the current remained constant for at least 10 min in darkness (p=1 at t = 12 min), and was restored upon illumination with 525 nm light. All values represent mean ± SEM.

DOI: https://doi.org/10.7554/eLife.37487.003

The following source data is available for figure 1:

**Source data 1.** Source data for *Figure 1E,F*.
DOI: https://doi.org/10.7554/eLife.37487.004

neurons (*Figure 2B*, *Figure 2—figure supplement 1A*). In contrast, expression was absent in the PPN and LDT (*Figure 2—figure supplement 1B*), in agreement with the lack of retrograde transport for lentiviruses (*Mazarakis et al., 2001*). Four to six weeks after viral infection, transduced coronal slices were treated with MAHoCh, and nicotine-induced currents were recorded from GFP-positive DA neurons. VTA DA neurons were identified based on their anatomical localization and electrophysiological properties, (i.e. pacemaker activity and typical action potential waveform), which are robust indicators of the DAergic signature (*Figure 2—figure supplement 2A–B*). Currents evoked by a local puff of nicotine were strongly inhibited under 380 nm light, and fully restored under 525 nm light (*Figure 2C*). Photo-inhibition was robust at both low and high concentrations of nicotine, and was absent in non-transduced slices treated with MAHoCh (*Figure 2D*). The degree of photo-inhibition was smaller than that observed in heterologous expression system, suggesting that only a subset of β2* receptors incorporated the cysteine-mutated β2. Importantly, over-expression of β2E61C did not significantly affect the amplitude of nicotine-induced currents (*Figure 2E*), indicating that the total number of functional nAChRs at the cell surface was unchanged. Moreover, MAHoCh alone had no detectable off-target effect on other endogenous ion channels or on resting or active membrane properties of the cell (*Figure 2—figure supplement 2C,D*), indicating that the effect of light was specific for β2E61C* nAChRs. Overall, these experiments show that β2E61C associates with endogenous nAChR subunits in DA neurons, to produce receptors with normal neurophysiological roles, while allowing specific photo-control of nicotinic signaling.

## β2*nAChRs control the firing patterns of VTA DA neurons

VTA DA neurons show two distinct patterns of electrical activity: tonic, regular-spiking in the low frequency range and transient sequences of high-frequency firing, referred to as bursts (*Paladini and Roeper, 2014*). Bursting activity, which is a crucial signal for behavioral conditioning (*Tsai et al., 2009*), is under the control of excitatory afferents from the PPN and LDT (*Lodge and Grace, 2006*; *Paladini and Roeper, 2014*; *Floresco et al., 2003*). We asked whether endogenous pontine ACh modulates the firing patterns of VTA DA neurons through β2*nAChRs. Testing this hypothesis required to deploy strategies for acutely manipulating nicotinic transmission in vivo, since DA neurons discharge only in pacemaker-like tonic activity in brain slices, due to cholinergic and glutamatergic afferents being severed (*Grace and Onn, 1989*). To this aim, we used a microdrive multielectrode manipulator (System mini matrix with five channels, *Figure 3A*) directly mounted onto the head of an anaesthetized mouse. This system allowed us to stereotaxically deliver the photoswitch and record the spontaneous activity of putative DA (pDA) neurons, while delivering alternating flashes of 390 and 520 nm light in the VTA (*Figure 3A,B*). β2E61C was virally transduced in the VTA of WT mice and recordings were performed three to four weeks after infection. MAHoCh was infused in the VTA at least an hour before starting the electrophysiological recordings, to allow the excess of untethered photoswitch to be cleared. We first found that the spontaneous activity of pDA neurons from WT and transduced animals were not significantly different in darkness (*Figure 3—figure supplement 1A*), indicating that viral expression of β2E61C did not affect the native physiology of the cells. We then checked whether alternatively switching light between 390 and 520 nm (20 cycles) affected the spontaneous firing of pDA neurons, by calculating the absolute percent of photoswitching (defined as the absolute value of (($Freq_{520}$ − $Freq_{390}$)/$Freq_{390}$)). Importantly, we found that switching wavelength impacted the spontaneous firing rate of MAHoCh-treated pDA

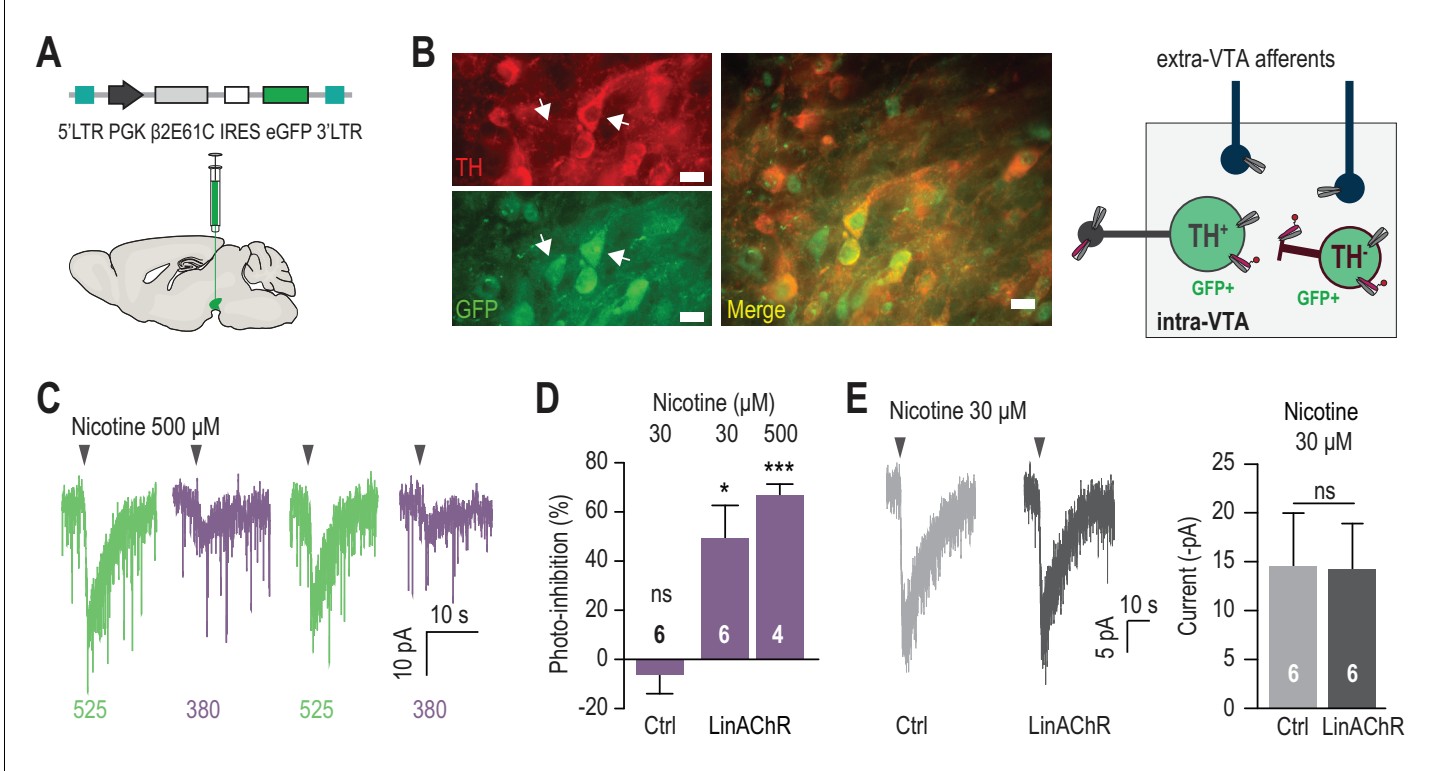

**Figure 2.** Reversible photo-inhibition of nAChR currents. (A) Viral transduction of the VTA using a lentivirus encoding pGK-β2E61C-IRES-eGFP. (B) Left, immunocytochemical identification of virally-transduced neurons (GFP-positive) 4 weeks after viral injection. DA neurons are labelled using anti-tyrosine hydroxylase (TH) antibodies. Note that the virus non-selectively transduces TH$^+$ and TH$^-$ neurons (arrows). Scale bar 10 μm. Right, scheme illustrating LinAChR expression profile. The β2E61C subunit (red) is incorporated into nAChRs on the soma, dendrites and axon terminals of TH$^+$ and TH$^-$ VTA neurons, but excluded from extra-VTA afferents. Local infusion of MAHoCh (red arm) into the VTA labels and photosensitizes solely intra-VTA receptors, and not receptors on DA terminals. (C) Representative photo-inhibition of nicotine-induced currents (500 μM, local puff 500 ms) recorded at −60 mV from a GFP-positive DA neuron labeled with MAHoCh (70 μM, 20 min) in an acute brain slice. (D) Average percent photo-inhibition of nicotinic currents (1-($I_{380}/I_{525}$)) evoked using a local puff (500 ms) of 30 μM (49.5 ± 13.2%, p=0.013, one sample t-test) or 500 μM nicotine (67.0 ± 4.3%, p=0.0006), recorded as in (C) from MAHoCh-treated GFP-positive DA neurons (n = 6 and 4 for nicotine 30 and 500 μM, respectively). Control neurons (MAHoCh alone, 30 μM nicotine, Ctrl) show no photo-inhibition (−6.3 ± 7.7%, p=0.453, n = 6). (E) Left: Representative currents induced by nicotine (30 μM) in a control neuron (Ctrl, grey) and a β2E61C-transduced neuron (LinAChR, black). Right: Control (n = 6) and transduced (n = 6) neurons display nicotine-induced currents of same amplitude (−14.5 ± 5.5 and −14.2 ± 4.7 pA, respectively, p=0.97). All values represent mean ± SEM.

DOI: https://doi.org/10.7554/eLife.37487.005

The following source data and figure supplements are available for figure 2:

**Source data 1.** Source data for *Figure 2D,E*.
DOI: https://doi.org/10.7554/eLife.37487.008

**Figure supplement 1.** Selective transduction of β2E61C in the VTA of WT mice.
DOI: https://doi.org/10.7554/eLife.37487.006

**Figure supplement 2.** No adverse effect of MAHoCh on the basic electrophysiological properties of WT VTA DA neurons.
DOI: https://doi.org/10.7554/eLife.37487.007

neurons of transduced animals, but not of control WT animals (*Figure 3C,D*), further evidencing that the effect of light is specific to the anchoring of MAHoCh to the β2 cysteine mutant.

For transduced animals, only a fraction of pDA neurons responded to light. To separately evaluate responding from non-responding neurons, we set a threshold (15% absolute photoswitching) to exclude 95% of the control neurons (*Figure 3D*). Based on this threshold, about a third (33/93) of the pDA neurons of transduced animals responded to light, compared to 1/28 for control animals. Non-responding neurons probably were either not transduced, or received too little endogenous cholinergic drive. We then compared the activity of each responding pDA neuron under both wavelengths of light and observed that some neurons responded with increased firing and some with decreased firing. A majority of the neurons (Type 1, 24/33) showed decreased activity under 390 nm

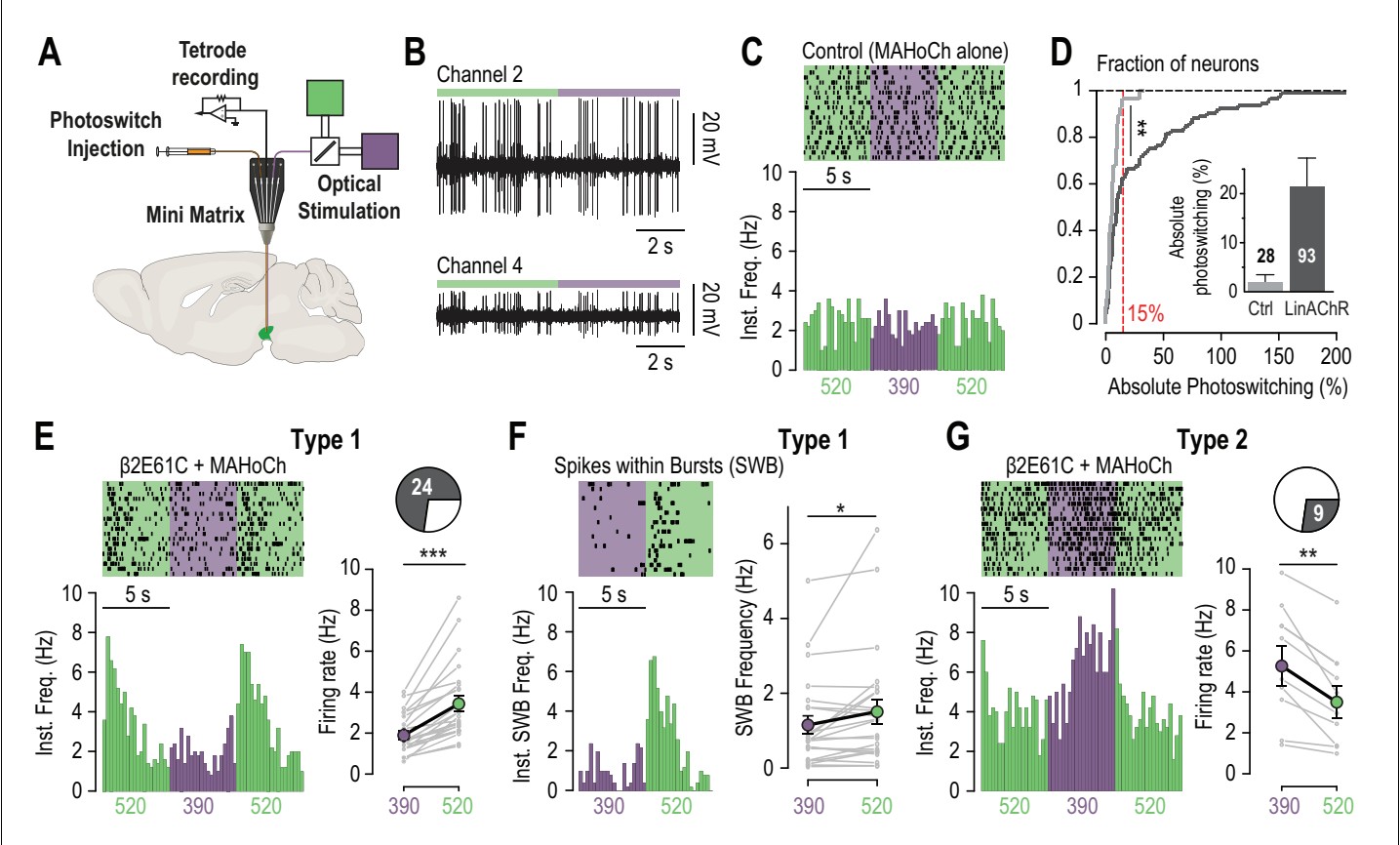

**Figure 3.** In vivo photo-control of endogenous cholinergic signaling. (**A**) Design of the experimental setup for concurrent recording and photocontrol of midbrain DA neurons in vivo. A micro-drive system (Mini Matrix) is mounted with a stereotaxic frame on the head of an anesthetized mouse, and enables to position in the VTA one cannula for photoswitch injection, up to three tetrodes for electrophysiological recordings, and one optic fiber connected to a beam combiner for optical stimulation. The photoswitch is injected at least an hour prior to the recordings. (**B**) Representative multi-unit recordings of transduced neurons on two channels of a tetrode while alternating illumination between 390 and 520 nm light. (**C**) Representative electrophysiological response of a MAHoCh-treated control neuron, while alternating illumination conditions between 390 (purple) and 520 nm light (green) every 5 s. Top, raster plot (n = 19 transitions) centered on the 390 nm light stimuli, and showing both the 390 to 520 and the 520 to 390 nm light transitions. Bottom, peri-stimulus time histogram (PSTH) of firing frequency using a 250 ms bin. (**D**) Change in firing frequency (expressed in absolute photoswitching) between 390 and 520 nm light for MAHoCh-treated control (Ctrl, light grey, n = 28) and β2E61C-transduced neurons (LinAChR, dark grey, n = 93). Photoswitching is calculated as (($Freq_{520}$ – $Freq_{390}$)/$Freq_{390}$) and represented in percent. Cumulative distribution indicates that virally transduced neurons significantly photoswitch compared to controls (p=0.0055, Kolmogorov-Smirnov test). Inset, absolute photoswitching for control neurons (1.87 ± 1.60%) is lower than that for transduced neurons (21.35 ± 5.90%). The threshold set at 15% absolute photoswitching (red) was used to determine the fraction of responding neurons in transduced animals (33/93, 35.5%). (**E**) Left, representative electrophysiological response of a virally transduced, MAHoCh-treated type 1 pDA neuron, represented as in (**C**). Right: Average firing rate of all type 1 pDA neurons (n = 24), under 520 (green) and 390 nm (purple) light. Firing frequency is significantly lower in 390 nm (1.85 Hz) compared to 520 nm light (3.41 Hz, p=1.19e$^{-07}$). (**F**) Top left, raster plot (n = 20 transitions) for the spikes contained within bursts (SWB) under 390 nm and 520 nm light, for the same neuron as in (**E**). Bottom left, PSTH of instantaneous SWB frequency using a 250 ms bin. Right, average SWB frequency of all type 1 pDA neurons (n = 24), under both wavelengths of light. SWB frequency is significantly lower in 390 compared to 520 nm light (p=0.043). (**G**) Left, representative electrophysiological response of a virally transduced, MAHoCh-treated type 2 pDA neuron, represented as in (**C**). Right: Average firing rate of all type 2 pDA neurons (n = 9), under 520 (green) and 390 nm (purple) light. Firing frequency is significantly higher in 390 nm (5.25 Hz) compared to 520 nm light (3.48 Hz, p=0.0039). All values represent mean ± SEM.

DOI: https://doi.org/10.7554/eLife.37487.009

The following source data and figure supplement are available for figure 3:

**Source data 1.** Source data for *Figure 3C,G*.

DOI: https://doi.org/10.7554/eLife.37487.011

**Figure supplement 1.** Photocontrolling VTA β2LinAChRs in vivo.

DOI: https://doi.org/10.7554/eLife.37487.010

(*Figure 3E*), and a transient increase upon switching back to 520 nm, consistent with a direct nAChRs antagonism on VTA DA neurons by *cis* MAHoCh and relief from antagonism when MAHoCh is switched to its *trans* state. The increase in firing upon relief from antagonism suggests that ambient ACh is sufficient to drive nAChRs in an activated state. In addition, bursting activity was significantly reduced in 390 nm light in Type 1 neurons, when β2*nAChRs were antagonized (*Figure 3F*). Hence, these receptors play a causal role in determining the firing patterns of VTA DA neurons. A smaller fraction of pDA neurons (Type 2, 9/33) showed the opposite profile, i.e. increased activity under 390 nm light compared to 520 nm (*Figure 3G*). This observation suggests that extracellular ACh acts on β2*nAChRs to exert an inhibitory drive on a sub-population of VTA DA neurons, possibly through an indirect network mechanism or through β2LinAChRs expressed on GABAergic interneurons. In Type 2 pDA neurons, we observed no effect of light on AP bursts (*Figure 3—figure supplement 1B*). Altogether, these results indicate that spontaneously-released ACh acts through post-synaptic β2*nAChRs (i.e. receptors expressed on intra-VTA neurons, see *Figure 2B*) to bi-directionally modulate the tonic firing and increase the bursting activity of VTA DA neurons. This excitatory/inhibitory nicotinic drive is consistent with the duality of the responses observed upon optogenetic activation of pontine cholinergic axons (*Dautan et al., 2016*), yet it directly implicates nicotinic- and not muscarinic- ACh receptors. It is also consistent with the concurrent excitations and inhibitions observed in DA neurons upon nicotine systemic injections (*Eddine et al., 2015*).

## Photo-controlling the effect of nicotine on VTA DA neurons

In WT mice, VTA DA neurons respond to nicotine with a rapid increase in firing frequency and in bursting activity, and these responses are totally absent in β2$^{-/-}$ mice (*Maskos et al., 2005*). Several pre- and post-synaptic mechanisms have been proposed to explain the effects of nicotine on DA cell firing (*Juarez and Han, 2016*; *Faure et al., 2014*). We tested whether blocking VTA β2LinAChRs resulted in a decrease response to nicotine in DA cells. To this aim, VTA DA neurons transduced with β2E61C were recorded in vivo using the juxta-cellular technique, which enables long, stable recordings and multiple drug injections (*Figure 4A,B*). Neurons that were successfully filled with neurobiotin (3 out of 7) were subsequently immuno-histologically identified as DAergic (*Figure 4—figure supplement 1A*). We found that the nicotine-induced variation in firing rate was much smaller under 390 nm light, when receptors were antagonized, and illumination with 520 nm light fully restored the initial response (*Figure 4C,D* and *Figure 4—figure supplement 1B*). Three of seven neurons tested showed spontaneous bursting, and all of these responded to nicotine by a variation in spikes within bursts (SWB) that appeared reduced under 390 nm light. Importantly, the response recorded from transduced animals was similar to that observed in WT animals (*Figure 4—figure supplement 1C,D*), further supporting the idea that the basic neurophysiological properties of DA neurons are unaffected by the viral transduction. Altogether, these experiments show that the effect of nicotine can be reversibly blocked with high spatial, temporal and pharmacological precision in defined brain structures, here the VTA.

## Blocking VTA nAChRs is sufficient to disrupt preference to nicotine

The VTA is crucial for the motivational properties of many drugs of abuse, including nicotine (*Di Chiara and Imperato, 1988*; *Volkow and Morales, 2015*). In rodents, nicotine increases the activity of VTA DA neurons (*Mameli-Engvall et al., 2006*; *Maskos et al., 2005*) and boosts DA release in the NAc (*Di Chiara and Imperato, 1988*), signaling its reinforcing, rewarding effect. We tested whether optically blocking β2*nAChRs of the VTA was sufficient to prevent nicotine from producing its reinforcing properties. To this aim, we chronically implanted above the transduced VTA a guide cannula for local delivery of the chemical photoswitch and light (*Figure 5A*) and subjected mice to a conditioned-place preference (CPP) protocol (*Figure 5B*). Proper transduction and placement of the cannula guide were confirmed immunohistochemically (*Figure 5—figure supplement 1A*). Consistent with previous reports (*Walters et al., 2006*), WT animals showed a significant place preference for nicotine while β2$^{-/-}$ mice did not (*Figure 5C* and *Figure 5—figure supplement 1B*). To determine whether nicotine preference could be reversibly photo-controlled in individual animals, CPP tests were conducted with two groups of β2E61C-transduced animals. Pairings were performed first with nicotine and 390 nm light for group 1, and with nicotine and 520 nm light for group 2. Two months after the first CPP test, nicotine pairing was performed with the alternative light condition,

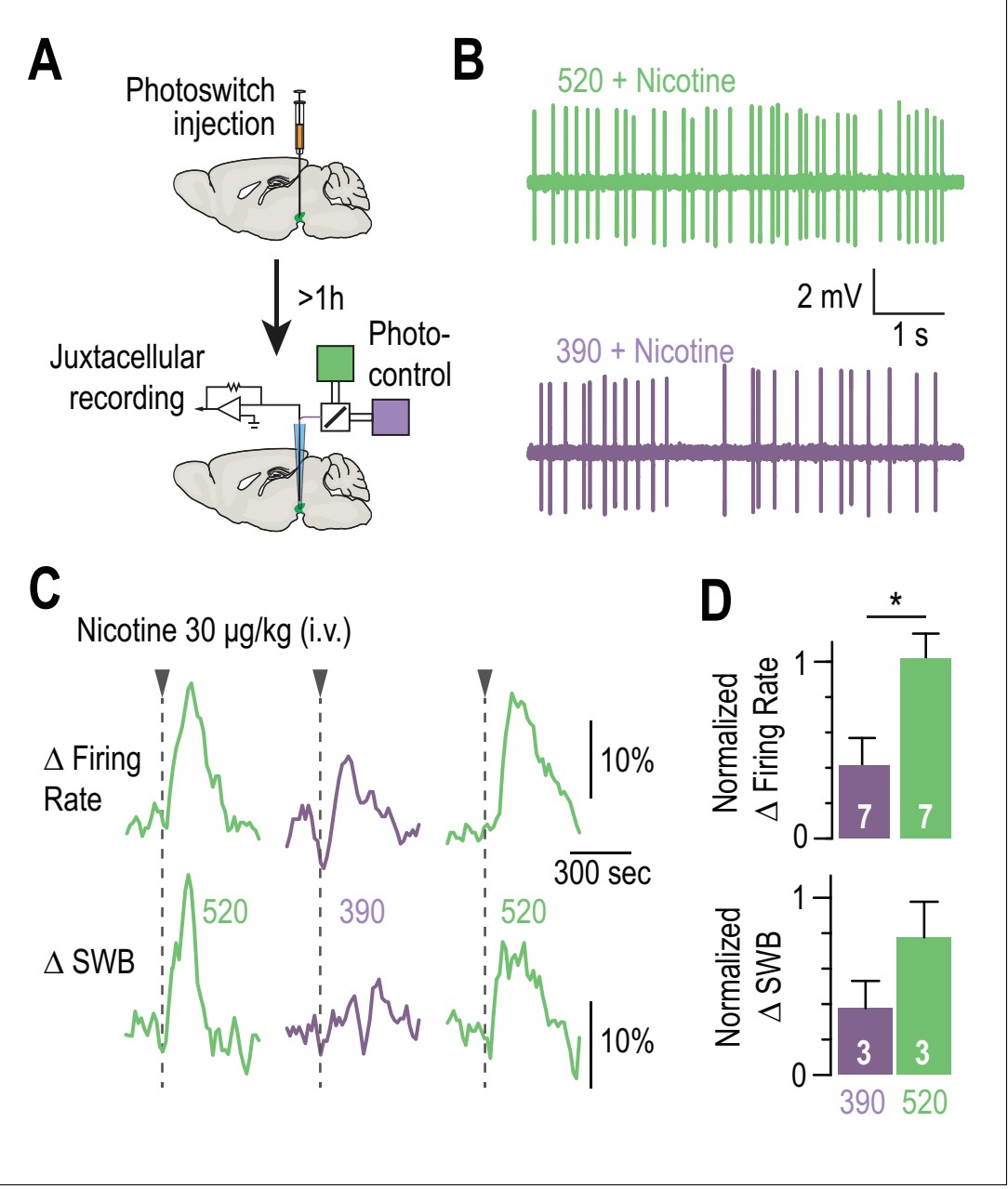

**Figure 4.** Blocking the effects of nicotine selectively in the VTA. (**A**) Experimental design for photoswitch injection and subsequent juxtacellular recording coupled to photocontrol. (**B**) Representative electrophysiological recording of one VTA DA neuron, during an i.v. injection of nicotine (30 µg/kg), under 520 (top, green) and 390 nm light (bottom, purple), showing greater electrical activity in green light. (**C**) Representative change in firing frequency (top) and in bursting activity (bottom) of a VTA DA neuron, elicited by an i.v. injection of nicotine (30 µg/kg), under 390 and 520 nm light, showing reversible photo-inhibition. (**D**) Top, average change in firing rate for VTA DA neurons (n = 7) upon nicotine injection under 390 (41.0 ± 15.7 %, purple) and 520 nm light (102.0 ± 14.0 %, green), normalized to the initial response in darkness. Change in firing frequency in 520 nm light is significantly different for 390 nm (p=0.015, Wilcoxon-Mann-Whitney test with Holm-Bonferroni correction) but not from darkness (p=0.81). Bottom, average change in SWB for bursting VTA DA neurons (n = 3) upon nicotine injection under 390 (37.7 ± 15.3 %, purple) and 520 nm light (77.5 ± 20.3 %, green), normalized to the initial response in darkness. All values represent mean ± SEM.

DOI: https://doi.org/10.7554/eLife.37487.012

The following source data and figure supplement are available for figure 4:

**Source data 1.** Source data for *Figure 4D*.

*Figure 4 continued on next page*

*Figure 4 continued*

DOI: https://doi.org/10.7554/eLife.37487.014

**Figure supplement 1.** The response of VTA DA neurons to nicotine is similar in WT and in β2E61C-transduced animals.

DOI: https://doi.org/10.7554/eLife.37487.013

i.e. 520 nm light for group 1 and 390 nm light for group 2. For both groups, animals showed preference to nicotine under 520 but not under 390 nm light (*Figure 5D,E*). These results cannot be attributed to changes in general activity behavior, since locomotion was not affected by viral transduction or light (*Figure 5—figure supplement 1C*). Altogether, these experiments show that nicotine-CPP can be reversibly switched on and off in the same animal, by manipulating β2*nAChRs selectively located in the VTA.

## Discussion

In this study, we used an optogenetic pharmacology strategy (*Kramer et al., 2013*) and demonstrated pharmacologically-specific, rapid local and reversible manipulation of brain nAChRs in behaving mice. Classical opsin-based optogenetics aims at turning specific neurons on or off for decoding neural circuits (*Kim et al., 2017*). Our strategy expands the optogenetic toolbox beyond excitation and inhibition by providing acute interruption of neurotransmission at the post-synaptic level, and provides mechanistic understanding of how specific transmitters and receptors contribute to modulation of circuits and behaviors.

Our method for photosensitizing receptors relies on the covalent attachment of a chemical photoswitch on a cysteine-modified receptor mutant. The photochemical properties of the azobenzene photoswitch make this strategy ideally suited for reversibly controlling neurotransmitter receptors

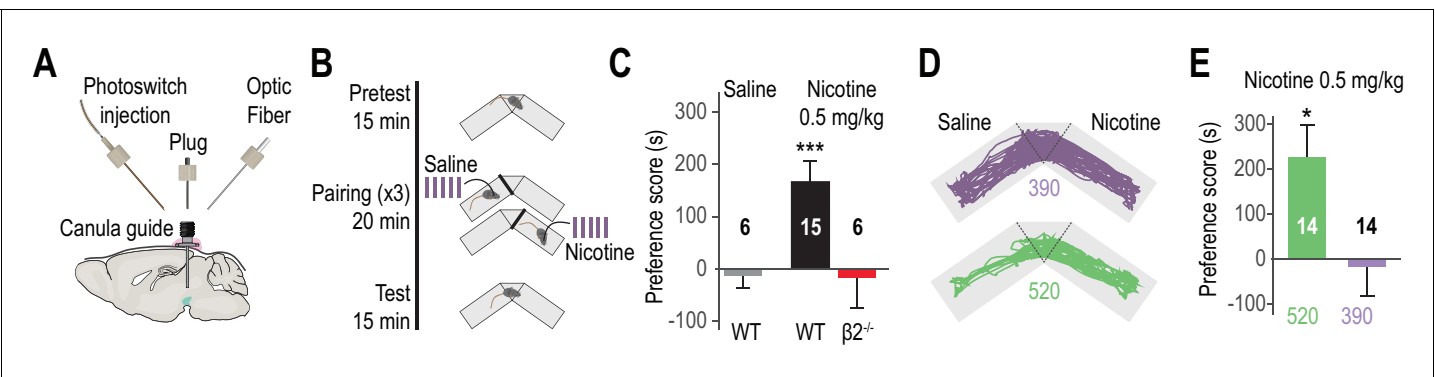

**Figure 5.** Reversibly disrupting nicotine preference. (**A**) Experimental design of the opto-fluidic device for opto-pharmacology experiments in freely-moving mice. The cannula guide is chronically implanted above the VTA and is used for both photoswitch and light delivery. (**B**) Nicotine-place preference protocol. Drug-free pretest (15 min) was followed by 3 consecutive days of pairing, which consisted in morning and evening saline and nicotine (0.5 mg/kg) conditioning sessions (20 min). For experiments using LinAChRs, mice were injected with the photoswitch in the morning and received light (390 or 520 nm, 2 s flashes at 0.1 Hz) in both pairing chambers. On day 5, mice were placed in the central chamber (no drug, no light) and were allowed to freely explore the environment. (**C**) Mean preference score (ps) for WT mice conditioned with saline (grey, n = 6, ps = −12.8 ± 24.0 s, p=0.69) and with nicotine (black, n = 15, ps = 165.9 ± 39.6 s, p=6.1e$^{-04}$), and for β2$^{-/-}$ mice conditioned with nicotine (red, n = 6, ps = −16.2 ± 58.7 s, p=0.44). (**D**) Representative trajectories of β2E61C-transduced and MAHoCh-treated mice conditioned with nicotine, under 390 (purple) and 520 nm light (green). (**E**) Mean preference for nicotine is abolished under 390 nm (purple, ps = −17.6 ± 63.8 s, p=0.80) and restored under 520 nm light (green, ps = 227.3 ± 72.1 s, p=0.015). Two groups of 7 mice were pooled. All values represent mean ± SEM.

DOI: https://doi.org/10.7554/eLife.37487.015

The following source data and figure supplement are available for figure 5:

**Source data 1.** Source data for *Figure 5C,E*.

DOI: https://doi.org/10.7554/eLife.37487.017

**Figure supplement 1.** Nicotine-induced CPP.

DOI: https://doi.org/10.7554/eLife.37487.016

with high efficacy and at speeds that rival synaptic transmission (*Lemoine et al., 2013*; *Levitz et al., 2013*; *Lin et al., 2015*; *Szobota et al., 2007*). Comparatively, strategies for photosensitizing proteins based on the fusion of light-sensitive modules (*Rost et al., 2017*) or chromophore-assisted light-inactivation (*Lin et al., 2013*; *Takemoto et al., 2017*) are too slow or irreversible, respectively. Due to the constrains of bioconjugation, in vivo use of photoswitch-tethered receptors in mice has been restricted to the eye (*Gaub et al., 2014*) and to superficial layers of the cerebral cortex (*Levitz et al., 2016*; *Lin et al., 2015*). Here, we demonstrate rapid on and off control of neuronal nAChRs in deep brain structures and in freely behaving animals. Our data show that photoswitch delivery resulted in an absolute subtype-specificity control of β2*nAChRs, with no apparent off-target effect. Labeling was rapid (minutes) and, due to its covalent nature, persisted for many hours (we have detected strong photosensitization in vivo up to 9 hr after treatment). Importantly, due to the thermal stability of MAHoCh, receptor function was unperturbed in darkness, while brief flashes of light were sufficient to bistably toggle LinAChR between its resting and antagonized states.

The cysteine-modified subunit was transduced in the VTA of WT mice. This resulted in a local replacement of the native β2 subunit with the cysteine-mutated version, while leaving nicotinic signaling in other brain regions (notably cholinergic pontine afferents) unaffected. Even though the WT β2 subunit remained in transduced cells, photoswitch treatment resulted in robust photo-sensitization of cysteine-mutated β2*nAChRs, indicating incorporation into heteromeric receptors. The pool of receptors remained apparently unchanged, most likely because endogenous nAChR subunits (e.g. α4) limit the total number of heteropentamers at the cell surface. Replacing the WT subunit by its cysteine counterpart in a knock-in animal would guarantee complete gene replacement and untouched expression profile. Yet, viral transduction affords the advantage of allowing the engineered receptor to be targeted for expression in specific types of neurons and in defined neuronal circuits. We used this feature to optically control nAChRs at the level of VTA neurons (both DAergic and non-DAergic cells, see *Figure 2B*), while leaving pre-synaptic receptors from various afferents unaffected, which would be impossible with a transgenic animal. Collectively, our results show that β2E61C competes with native subunits to form functional receptors that, once labeled with MAHoCh, retain their natural functions in darkness, and are made photo-controllable.

Cholinergic neurons from PPN and LDT project extensively to the VTA and substantia nigra (*Beier et al., 2015*) and are thought to form connections with downstream DAergic and GABAergic neurons through non-synaptic volume transmission. Optogenetic activation of cholinergic pontine axons induces post-synaptic currents in VTA DA neurons that have both nicotinic and glutamatergic signatures (*Xiao et al., 2016*), suggesting that extracellular ACh potentiates glutamate release by activating nAChRs located on axon terminals. Contrasting with this view, we show here that activation of post-synaptic (i.e. from intra-site) β2 nAChRs by endogenous ACh is sufficient to fine tune both the tonic and burst firing modes of VTA DA neurons. Furthermore, our results add temporal and causal considerations to previous genetic studies (*Mameli-Engvall et al., 2006*; *Tolu et al., 2013*) by establishing a direct relationship between the activity of β2 nAChRs and the firing patterns of VTA DA neurons. The rebound activity that occurred within 500 ms after de-antagonizing LinAChRs indeed suggests that, even though cholinergic inputs to the VTA are considered sparse, the extracellular levels of ACh are sufficient to activate a large population of receptors and greatly modify the electrical activity of DA neurons. Moreover, we identified a sub-population of VTA DA neurons that is inhibited when β2 nAChRs are de-antagonized, which suggests multiple functional mechanisms by which the cholinergic brainstem neurons may influence the activity of midbrain DA neurons. These results are coherent with the growing body of evidence that show that VTA DA neurons are heterogeneous in their physiological properties (*Morales and Margolis, 2017*; *Yang et al., 2018*) and in their responses to drugs (*Juarez and Han, 2016*), including nicotine (*Eddine et al., 2015*).

The rewarding properties of nicotine, and especially reinforcement during the acquisition phase of addiction, implicate an elevation of DA in the NAc (*Di Chiara and Imperato, 1988*). Nicotine administration directly depolarizes and activates VTA DA neurons and, consequently, increases extracellular striatal DA (*Maskos et al., 2005*; *Tolu et al., 2013*). Nicotine can also increase DA neuron firing by acting on GABAergic and glutamatergic afferent terminals, from local interneurons and projection fibers (*Mansvelder et al., 2002*). Finally, nicotine also modulates DA release by desensitizing nAChRs expressed in the striatum at the level of DA terminals (*Rice and Cragg, 2004*). These different studies suggest alternative circuit mechanisms to explain the outcome of nicotine action on

VTA circuitry, for reviews see (*Juarez and Han, 2016*; *Faure et al., 2014*). We took advantage of the anatomical and cellular resolution of our approach, and locally blocked the effect of nicotine on VTA DA and non-DA neurons, while leaving pre-synaptic receptors of afferents from other brain areas and of striatal DA terminals unaffected. Our results show that β2*nAChRs of VTA neurons are a key player of both the response to nicotine at the cellular level, and the rewarding properties of this addictive substance at the behavioral level. Importantly, blocking the excitatory phasic input produced by nicotine was sufficient to completely prevent reinforcement learning. This is consistent with our results concerning the ability of β2 nAChRs to tune burst firing in DA neurons, and with the fact that activation of LDT-to-VTA cholinergic neurons causes positive reinforcement (*Dautan et al., 2016*; *Xiao et al., 2016*). All together, these results strongly suggest that these receptors have a central role in reward processing.

There is a considerable interest to target-specific nAChRs and specific circuits to treat psychiatric disorders such as addiction, depression or schizophrenia. Yet, we do not know which native receptor subtype mediates specific physiological or pathological function, hampering development of clinically effective drugs, notably for preventing or treating addiction. Optogenetic pharmacology offers the unique opportunity to locally and reversibly 'knock-out' the function of a specific receptor isoform in vivo, and to directly evaluate within the same animal the consequences at the cellular, circuit and behavioral levels. Our approach should be applicable to other photo-activatable and -inhibitable nAChR subtypes and other neuronal circuits, and may provide a platform for examining new translational strategies for treating neuropsychiatric disorders.

## Materials and methods

### Key resources table

| Reagent type (species) or resource | Designation | Source or reference | Identifiers | Additional information |
|---|---|---|---|---|
| Antibody | Anti-tyrosine Hydroxylase produced in mouse | Sigma-Aldrich | T1299, RRID:AB_477560 | |
| Antibody | Anti-Choline-Acetyltransferase produced in goat | Merck-Millipore | AB144, RRID:AB_90650 | |
| Antibody | Anti-GFP produced in rabbit | | | |
| Antibody | Anti-GFP produced in chicken | Aves Lab | GFP-1020, RRID:AB_10000240 | |
| Antibody | Anti-rabbit Cy2-conjugated produced in donkey | Jackson Immuno Research | 711-225-152, RRID:AB_2340612 | |
| Antibody | Anti-mouse Cy3-conjugated produced in donkey | Jackson Immuno Research | 715-165-150, RRID:AB_2340813 | |
| Antibody | Anti-chicken Alexa488-conjugated | Jackson Immuno Research | 703-545-155, RRID:AB_2340375 | |
| Antibody | anti-goat Alexa 555-conjugated produced in donkey | Life Technologies | A21432, RRID:AB_141788 | |
| Antibody | AMCA-Streptavidin | Jackson ImmunoResearch | 016-150-084, RRID:AB_2337243 | |
| Strain, strain background (*mus musculus*, males) | C57Bl/6JRj | Janvier Laboratories, France | SC-C57J-M, RRID:MGI:5752053 | |

*Continued on next page*

*Continued*

| Reagent type (species) or resource | Designation | Source or reference | Identifiers | Additional information |
|---|---|---|---|---|
| Strain, strain background (*mus musculus*, males) | ACNB2 | https://doi.org/ 10.1038/374065a0 | | maintained on a C57BL6/J background |
| Strain, strain background (*lentivirus*) | Lenti-pGK-B2E61C-IRES-GFP | This paper | | |
| Cell line (*mus musculus*) | Neuro 2a | Sigma-Aldrich | 89121404-1VL, RRID:CVCL_0470 | |
| Transfected construct (*mus musculus*) | pIRES-a4-IRES-GFP | https://doi.org/10.1038/ nature03694 | | |
| Transfected construct (*mus musculus*) | pIRES-b2E61C-IRES-eGFP | this paper | | |
| Chemical compound, drug | MAHoCh | https://doi.org/10.1038/ nchem.1234 | | |
| Chemical compound, drug | NaCl | Sigma-Aldrich | S7653 | |
| Chemical compound, drug | KCl | Sigma-Aldrich | P9333 | |
| Chemical compound, drug | NaH2PO4 | Sigma-Aldrich | S8282 | |
| Chemical compound, drug | MgCl2 | Sigma-Aldrich | M2670 | |
| Chemical compound, drug | CaCl2 | Sigma-Aldrich | 233506 | |
| Chemical compound, drug | NaHCO3 | Sigma-Aldrich | S6297 | |
| Chemical compound, drug | Sucrose | Sigma-Aldrich | S0389 | |
| Chemical compound, drug | Glucose | Sigma-Aldrich | 49159 | |
| Chemical compound, drug | Kynurenic Acid | Sigma-Aldrich | K3375 | |
| Chemical compound, drug | Albumin, from bovine serum | Sigma-Aldrich | A4503 | |
| Chemical compound, drug | KGlu | Sigma-Aldrich | P1847 | |
| Chemical compound, drug | HEPES | Sigma-Aldrich | H3375 | |
| Chemical compound, drug | EGTA | Sigma-Aldrich | E3889 | |

*Continued on next page*

*Continued*

| Reagent type (species) or resource | Designation | Source or reference | Identifiers | Additional information |
|---|---|---|---|---|
| Chemical compound, drug | ATP | Sigma-Aldrich | A9187 | |
| Chemical compound, drug | GTP | Sigma-Aldrich | G8877 | |
| Chemical compound, drug | Biocytin | Sigma-Aldrich | B4261 | |
| Chemical compound, drug | Nicotine tartrate | Sigma-Aldrich | N5260 | |
| Chemical compound, drug | Glucose | Sigma-Aldrich | G8270 | |
| Chemical compound, drug | DMEM + Glutamax | Life Technologies | 31966–021 | |
| Chemical compound, drug | FBS | Life Technologies | 10500–064 | |
| Chemical compound, drug | Non-essential amino acids | Life Technologies | 11140–035 | |
| Chemical compound, drug | Pennicilin/ Streptomycin | Life Technologies | 15140–122 | |
| Chemical compound, drug | Trypsin | Life Technologies | 15090–046 | |
| Chemical compound, drug | Polylysine | Sigma-Aldrich | P6282 | |
| Chemical compound, drug | DMSO | Sigma-Aldrich | D2650 | |
| Chemical compound, drug | Carbamylcholine Chloride | Sigma-Aldrich | C4382 | |
| Chemical compound, drug | DPBS 10x | Life Technologies | 14200–067 | |
| Chemical compound, drug | Neurobiotin Tracer | Vector laboratories | SP-1120 | |
| Chemical compound, drug | Prolong Gold Antifade Reagent | Invitrogen | P36930 | |
| Software, algorithm | MATLAB | MathWorks | RRID:SCR_001622 | |
| Software, algorithm | R Project for Statistical Computing | http://www.r-project.org/ | RRID:SCR_001905 | |
| Software, algorithm | Fiji | http://fiji.sc | RRID:SCR_002285 | |

*Continued on next page*

*Continued*

| Reagent type (species) or resource | Designation | Source or reference | Identifiers | Additional information |
|---|---|---|---|---|
| Software, algorithm | Adobe Illustrator CS6 | Adobe | RRID:SCR_010279 | |
| Software, algorithm | Clampfit (pClamp suite) | Molecular Devices | RRID:SCR_011323 | |

## Animals

65 Wild-type male C57BL/6J mice were obtained from Janvier Laboratories (France) and 6 knockout SOPF-HO-ACNB2 ($\beta2^{-/-}$) male mice were obtained from Charles Rivers Laboratories (France). $\beta2^{-/-}$ mice were generated as described previously (*Picciotto et al., 1995*). Even though WT and $\beta2^{-/-}$ mice are not littermates the mutant line was generated more than 20 years ago, and has been back-crossed more than 20 generations with the WT C57BL/6J line and is more than 99.99% C57BL/6J. All experiments were performed on mice between 8 and 16 weeks of age. All experiments were performed in accordance with the recommendations for animal experiments issued by the European Commission directives 219/1990, 220/1990 and 2010/63, and approved by Sorbonne Université.

## Chemical photoswitch

MAHoCh was synthesized as described previously (*Tochitsky et al., 2012*) and was stored as concentrated stock solutions (100 mM) in water-free DMSO at $-80°C$. For cell labeling, aqueous solutions of MAHoCh were prepared extemporaneously.

## Light intensity measurements

Light intensities were measured with a power meter (1916 R, Newport) equipped with a UV-silicon wand detector (818-ST2-DB Newport).

## Molecular biology and virus production

The cDNAs for the WT mouse $\beta2$ and $\alpha4$ nAChR subunits were from previously-designed pIRES (CMV promoter) or pLenti (pGK promoter) vectors (*Maskos et al., 2005*). All the constructs are bi-cistronic, with an IRES-eGFP sequence designed to express eGFP and the nAChR subunit using the same promoter. The pLenti construct also contains the long terminal repeats, WPRE and virus elements for packaging into lentiviral vectors. The single cysteine mutation E61C was inserted into pIRES-CMV-$\beta2$-IRES-eGFP and pLenti-pGK-$\beta2$-IRES-eGFP by site-directed mutagenesis using the Quickchange II XL kit (Agilent). Mutations were verified by DNA sequencing. Lentiviruses were prepared as described previously (*Maskos et al., 2005*) with a titer of 150 ng of p24 protein in 2 µl.

## Cell line

We used Neuro2A cells (Sigma Aldrich #89121404-1VL), a mouse neuroblastoma cell line classically used for nAChRs expression (*Xiao et al., 2011*). Cells were certified by Sigma-Aldrich. Mycoplasma contamination status were negative.

## Cell culture, transfection and labeling

Briefly, Neuro2A cells were cultured in Dulbecco's Modified Eagle's Medium (DMEM), supplemented with 10% Foetal Bovine Serum (FBS), 1% non-essential amino-acids, 100 units/ml penicillin, 100 mg/ml streptomycin and 2 mM glutamax in a 5% $CO_2$ incubator at 37°C. Cells were transfected overnight with a 1:1 ratio of $\alpha4$ and $\beta2E61C$ subunits (pLenti-pGK-$\alpha4$-IRES-eGFP and pLenti-pGK-$\beta2E61C$-IRES-eGFP), using calcium-phosphate transfection method (*Lemoine et al., 2016*). Cells were used 2–3 days after transfection for electrophysiology. Prior to recordings, cells were labeled with MAHoCh (20 µM in external solution) for 20 min.

## Stereotaxic viral injections

WT mice (6–8 weeks) were anaesthetized with 1% isoflurane gas and placed in a stereotaxic frame (David Kopf). A small craniotomy was made above the location of the VTA. A lentivirus containing the construct pGK-$\beta2E61C$-IRES-eGFP was injected in the VTA (1 µl at the rate of 0.1 µl/min) with a

10 µl syringe (Hamilton) coupled with a polyethylene tubing to a 36 G cannula (Phymep), with the following coordinates [AP: −3.1 mm; ML:±0.4 mm; DV: −4.7 mm from bregma]. Mice were then housed during at least 4 weeks before electrophysiology or behavior experiments.

## Midbrain slices preparation and labeling

4–8 weeks after viral infection, mice were deeply anesthetized with an i.p. injection of a mixture of ketamine (150 mg/kg, Imalgene 1000, Merial) and xylazine (60 mg/kg, Rompun 2%, Bayer). Coronal midbrain sections (250 µm) were sliced using a Compresstome (VF-200; Precisionary Instruments) after intra-cardiac perfusion of cold (0–4°C) sucrose-based artificial cerebrospinal fluid (SB-aCSF) containing (in mM): 125 NaCl, 2.5 KCl, 1.25 $NaH_2PO_4$, 5.9 $MgCl_2$, 26 $NaHCO_3$, 25 Sucrose, 2.5 Glucose, 1 Kynurenate. After 10 min at 35°C for recovery, slices were transferred into oxygenated (95% $CO_2$/ 5% $O_2$) aCSF containing (in mM): 125 NaCl, 2.5 KCl, 1.25 $NaH_2PO_4$, 2 $CaCl_2$, 1 $MgCl_2$, 26 $NaHCO_3$, 15 Sucrose, 10 Glucose at room temperature for the rest of the day. Slices were labeled individually with MAHoCh (70 µM) in oxygenated aCSF (1 ml) for 20 min, and transferred to a recording chamber continuously perfused at 2 ml/min with oxygenated aCSF.

## Patch-clamp recordings

Patch pipettes (5–8 MΩ) were pulled from thin wall borosilicate glass (G150TF-3, Warner Instruments) using a micropipette puller (P-87, Sutter Instruments) and filled with a K-Gluconate based intra-pipette solution containing (in mM): 116 KGlu, 20 HEPES, 0.5 EGTA, 6 KCl, 2 NaCl, 4 ATP, 0.3 GTP and 2 mg/mL biocytin (pH adjusted to 7.2). Cells were visualized using an upright microscope with a Dodt contrast lens and illuminated with a white light source (Scientifica). A 460 nm LED (pE-2, Cooled) was used for visualizing eGFP positive cells (using a bandpass filter cube, AHF). Optical stimulation was applied through the microscope with two LEDs (380 and 525 nm, pE-2, CoolLED), with a light output of 6.5 and 15 mW, corresponding to 5 and 11.7 mW/mm$^2$ at the focal plane, respectively. Whole-cell recordings were performed using a patch-clamp amplifier (Axoclamp 200B, Molecular Devices) connected to a Digidata (1550 LowNoise acquisition system, Molecular Devices). Currents were recorded in voltage-clamp mode at −60 mV. Signals were low pass filtered (Bessel, 2 kHz) and collected at 10 kHz using the data acquisition software pClamp 10.5 (Molecular Devices). Electrophysiological recordings were extracted using Clampfit (Molecular Devices) and analyzed with R.

To record nicotinic currents from GFP-positive Neuro2A cells, we used the following external solution (containing in mM): 140 NaCl, 2.8 KCl, 2 $CaCl_2$, 2 $MgCl_2$, 10 HEPES, 12 glucose (pH 7.3 with NaOH). We used a computer-controlled, fast-perfusion stepper system (SF-77B, Harvard Apparatus) to apply nicotine-tartrate (100 µM, Sigma-Aldrich) or carbamylcholine chloride (CCh, 1 mM, Sigma-Aldrich), with an interval of 2 min, under different light conditions.

To record nicotinic currents from VTA DA neurons, local puffs (500 ms) of nicotine tartrate (30–500 µM in aCSF) were applied every minute, while alternating wavelengths, using a glass pipette (2–3 µm diameter) positioned 20 to 30 µm away from the soma and connected to a picospritzer (World Precision Instruments, adjusted to ~2 psi). DA neurons were characterized in current clamp mode as described in (*Lammel et al., 2008*), see *Figure 2—figure supplement 2A*. In some instances, at the end of the recording, the pipette was retracted carefully to allow labeling of the neuron with biocytin (*Marx et al., 2012*).

## In vivo juxtacellular recordings

4–8 weeks after viral infection, mice were deeply anaesthetized with chloral hydrate (8%, 400 mg/kg i.p.), supplemented as required to maintain optimal anesthesia throughout the experimental day. The scalp was opened and a hole was drilled in the skull above the location of the VTA. The saphenous vein was catheterized for intravenous administration of nicotine. Prior to recordings (at least 1 hr), 500 nl of a 400 µM solution of MAHoCh in aCSF were injected within the VTA at a rate of 50 nl/ min. Extracellular recording electrodes were made from 1.5 mm O.D./1.17 mm I.D. borosilicate glass (Harvard Apparatus) using a vertical electrode puller (Narishige). Under a microscope, the tip was broken to obtain a diameter of 1–2 µm. The electrodes were filled with a 0.5% Na-Acetate solution containing 1.5% of neurobiotin tracer yielding impedances of 20–50 MΩ. Electrophysiological signals were amplified with a headstage (1x, Axon Instruments) coupled to a high-impedance amplifier

(Axoclamp-2A, Axon Instruments) and audio monitored (A.M. Systems Inc.). The signal was digitized (Micro-2, Cambridge Electronic Design), sampled at 12.5 kHz and recorded using Spike2 software (CED). DA neurons were sampled in the VTA with the following coordinates: [AP: −3 to −4 mm; ML: +0.3 to+0.6 mm; DV: −4 to −4.8 mm, from Bregma]. Spontaneously active pDA neurons were identified on the basis of previously established electrophysiological criteria: 1) regular firing rate; 2) firing frequency between 1 and 10 Hz; 3) half AP >1.1 ms. After a baseline recording of at least 5 min, a saline solution (0.9% sodium chloride) was injected into the saphenous vein, and after another 10 min, injections of nicotine- tartrate (30 µg/kg) were administered via the same route in a final volume of 10 µl and under different light conditions (Dark – 390 nm – 520 nm). Successive injections (up to 6) were performed after the neuron returned to its baseline, or when the firing activity returned stable for at least 3 min. Light was applied through an optical fiber (500 µm core, NA = 0.5, Prizmatix) inserted within the glass pipette electrode and coupled through a combiner to 390/520 nm ultra-high-power LEDs (Prizmatix), yielding an output intensity of 4–8 mW at the tip of the fiber for each wavelength. Light was TTL-controlled and applied 10 s before nicotine injection, for 30 s total. When possible, neurons were electroporated and neurobiotin was expulsed from the electrode using positive current pulses as already described (*Pinault, 1996*; *Eddine et al., 2015*). Spikes Within Bursts (SWB) were identified as a sequence of spikes with the following features: (1) short intervals, (2) progressively decreasing spike amplitude, and (3) a progressively increasing inter-spike interval (ISI). When considering extracellular recordings, most studies use two criteria to automatically detect bursts: (1) their onset are defined by two consecutive spikes with an interval inferior to 80 ms, whenever (2) they are closed with an interval greater than 160 ms (*Grace and Bunney, 1984*). Firing rate and %SWB were measured on successive windows of 60 s, with a 45 s overlapping period. Responses to nicotine are presented as the mean percentage of firing frequency or %SWB variation from the baseline ±SEM. For photoswitching, maximum of firing variation induced by nicotine occurring 200 s after the injection in purple and green was normalized to the maximum of firing variation in darkness. Spikes were extracted with Spike2 (CED) and analyzed with R (https://www.r-project. org).

## In vivo multi-unit extracellular recordings

4–8 weeks after viral infection, mice were deeply anaesthetized with chloral hydrate (8%, 400 mg/kg i.p.), supplemented as required to maintain optimal anesthesia throughout the experiment. The scalp was opened and a hole was drilled in the skull above the location of the VTA. We used a Mini-Matrix (*Figure 3A*, Thomas Recording) allowing us to lower within the VTA: up to 3 tetrodes (Tip shape A, Thomas Recording, Z = 1–2 MΩ), a stainless-steel cannula (OD 120 µm, Thomas Recording) for photoswitch injection and a tip-shaped quartz optical fiber (100 µm core, NA = 0.22, Thomas Recording) for photostimulation. The fiber was coupled to a 390/520 nm LED combiner (Prizmatix) with an output intensity of 200–500 µW at the tip of the fiber for both wavelengths. These five elements could be moved independently with micrometer precision. 500 nl of MAHoCh (400 µM in aCSF) were infused (rate: 1 nl/s) within the VTA, and tetrodes were subsequently lowered in the same zone to record neurons. Spontaneously active pDA neurons were recorded at least 30 min after MAHoCh infusion and were identified on the basis of the electrophysiological criteria used for juxtacellular recordings. The optical fiber was then lowered 100–200 µm above the tetrodes. Baseline activity was recorded for 200 s in darkness, prior to applying 5 s light flashes of alternative wavelengths (390 nm / 520 nm). Electrophysiological signals were acquired with a 20 channels preamplifier included in the Mini Matrix (Thomas Recording) connected to an amplifier (Digital Lynx SX 32 channels, Neuralynx) digitized and recorded using Cheetah software (Neuralynx). Spikes were detected using a custom-written Matlab routine and sorted using a classical principal component analysis associated with a cluster cutting method (SpikeSort3D Software, Neuralynx). Neurons were considered as responding when their change in firing rate (% Photoswitching) at the transition from violet to green light exceeded a threshold of 15%, defined as the maximal % photoswitching observed in controls. This threshold was used for all recorded neurons in every condition. To extract the spikes contained within bursting episodes (SWB) we used the same criteria described in the juxtacellular recordings section. They are represented as the frequency of SWB because of the short analysis window (5 s). All the data were analyzed with R (https://www.r-project.org) and Matlab (MathWorks).

## Chronic guide cannula implantation

Following stereotaxic viral infection in the VTA (as described above), mice were implanted with a chronic opto-fluid guide cannula (Doric Lenses Inc, Canada, see *Figure 5A*) using the same coordinates. This guide (length = 3 mm from skull surface, ID/OD = 320/430 µm) has interchangeable threaded connectors and is used either with a fluid injection needle (protruding to 4.8 mm from skull surface) for delivering MAHoCh, or with an optic fiber injector (240 µm core, NA = 0.63, protruding to 4.8 mm from skull surface) coupled to a ceramic ferrule (1.25 mm) for light delivery. In-between experiments, a plug is used to close the guide cannula and thus seal the implant. The implant is attached to the skull with a dental cement (SuperBond, Sun Medical).

## Nicotine-induced place preference paradigm

The Conditioned Place Preference (CPP) box (Imetronic, France) consists of a Y-maze with one closed arm, and two other arms with manually operated doors. Two rectangular chambers (11 × 25 cm) with different cues (texture and color), are separated by a center triangular compartment (side of 11 cm). One pairing compartment has grey textured floor and walls and the other one has smooth black and white striped walls and floor. The first day (pretest) of the experiment, mice (n = 6–8 animals/group) explored the environment for 900 s (15 min) and the time spent in each compartment was recorded. Pretest data were used to segregate the animals with equal bias so each group has an initial preference score almost null, indicating no preference on average. On day 2, 3 and 4, animals received an i.p. injection of nicotine tartrate (0.5 mg/kg, in PBS) or an equivalent injection of saline (PBS), and immediately confined to one of the pairing chamber for 1200 s (20 min). The CPP test was performed using a single nicotine concentration (0.5 mg/kg) which is known to induce preference in mice (*Walters et al., 2006*). Groups were balanced so the animals do not always get nicotine in the same chamber. On the evening of the same day, mice received an injection of the alternate solution (nicotine or saline) and were placed in the opposite pairing chamber. The saline control animals received a saline injection in both pairing compartments. On day 5 (test), animals were allowed to explore the whole open-field for 900 s (15 min), and the time spent in each chamber was recorded. The preference score (ps) is expressed in seconds and is calculated by subtracting pretest from test data. Trajectories and time spent on each side are calculated based upon animal detection. Place preference and locomotor activity were recorded using a video camera, connected to a video-track system, out of sight of the experimenter. A home-made software (Labview 2014, National Instruments) tracked the animal, recorded its trajectory (20 frames per s) for 15 min and sent TTL pulses to the LED controller when appropriate (pairing sessions). For optogenetic pharmacology experiments, MAHoCh (400 µM in aCSF, 500 nl in 5 min) was injected early in the morning of pairing days (2, 3 and 4) under light gas anesthesia (Isoflurane 1%). 520/390 nm light was applied during pairing sessions (day 2, 3 and 4), on both sides, through a patch cord (500 µm core, NA = 0.5, Prizmatix, Israel) connected to the implanted ferrule with a sleeve and to the 390/520 nm combined UHP-LEDs (Prizmatix). Light was applied with the following pattern: 2 s pulses à 0.1 Hz with a measured output intensity of 10 mW at the tip of the patch cord. Light was not applied during pre-test and test. Behavioral data were collected and analyzed using home-made LabVIEW (National Instruments) and Matlab (MathWorks) routines.

## Immunohistochemistry

After patch-clamp experiments, individual slices (250 µm) were transferred in 4% paraformaldehyde (PFA) for 12–24 hr and then to PBS, and kept at 4°C. At the end of in vivo experiments, transduced mice received, under deep anesthesia (Ketamine/Xylazine), an intra-cardiac perfusion of (1) PBS (50 ml) and (2) paraformaldehyde (4% PFA, 50 ml) and brains were rapidly removed and let in 4% PFA for 48–72 hr of fixation at 4°C. Serial 60 µm sections of the ROI were cut with a vibratome. Immunohistochemistry was performed as follows: Floating VTA brain sections were incubated 1 hr at 4°C in a solution of phosphate-buffered saline (PBS) containing 3% Bovine Serum Albumin (BSA, Sigma; A4503) and 0.2% Triton X-100 and then incubated overnight at 4°C with a mouse anti-Tyrosine Hydroxylase antibody (TH, Sigma, T1299) at 1:200 dilution and a rabbit anti-GFP antibody (Molecular Probes, A-6455) at 1:500 dilution in PBS containing 1.5% BSA and 0.2% Triton X-100. The following day, sections were rinsed with PBS and then incubated 3 hr at 22–25°C with Cy3-conjugated anti-mouse and Cy2-conjugated anti-rabbit secondary antibodies (Jackson ImmunoResearch, 715-165-

150 and 711-225-152) at 1:200 and 1:1000 dilution respectively in a solution of 1.5% BSA and 0.2% Triton X-100 in PBS. In the case of biocytin/neurobiotin labelling, TH identification of the neuron was performed using AMCA-conjugated Streptavidin (Jackson ImmunoResearch) at 1:200 dilution. Floating pons sections were incubated 1 hr at 4°C in a solution of phosphate-buffered saline containing 0.2% Gelatine from cold-water fish skin (Sigma; G7041) and 0.25% Triton X-100 (PBS-GT) and then incubated overnight at 4°C a goat anti-Choline Acetyl-Transferase antibody (ChAT, Merck-Millipore, AB144) at 1:200 dilution and a chicken anti-GFP antibody (Aves Lab, GFP-1020) at 1:500 dilution in PBS-GT. The following day, sections were rinsed with PBS and then incubated 3 hr at 22–25°C with a donkey anti-goat Alexa 555-conjugated (Invitrogen, A21432) and donkey anti-chicken Alexa 488-conjugated (Jackson ImmunoResearch, 703-545-155) at 1:200 and 1:1000 dilution respectively in a solution of PBS-GT. After three rinses in PBS (5 min), wet slices were mounted using Prolong Gold Antifade Reagent (Invitrogen, P36930). Microscopy was carried out either with a confocal microscope (Leica) or an epifluorescence microscope (Leica), and images captured using a camera and analyzed with ImageJ software.

## Statistical analysis

No statistical methods were used to predetermine sample sizes. Data are plotted as mean ±SEM. Total number (n) of observations in each group and statistics used are indicated in figure and/or figure legend. Unless otherwise stated, comparisons between means were performed using parametric tests (two-sample t-test) when parameters followed a normal distribution (Shapiro test $p>0.05$), and non-parametric tests (here, Wilcoxon or Mann-Whitney (U-test)) when this was not the case. Homogeneity of variances was tested preliminarily and the t-tests were Welch-corrected if needed. Multiple comparisons were Holm-Bonferroni corrected. Comparison between the cumulative distributions of in vivo multi-unit recordings between controls and LinAChRs (*Figure 3D*) was performed using a Kolmogorov-Smirnov test. $p>0.05$ was considered to be not statistically significant.

## Acknowledgments

Authors would like to thank Nadine Mouttajagane, Ambre Bonnet and Michael Martin for molecular biology work, and Justine Hadjerci, Steve Didienne and Samir Takillah for their help with electrophysiology and behavior experiments. This work was supported by the Agence Nationale de la Recherche (ANR-JCJC 2014 to A.M.), by a NARSAD Young Investigator Grant from the Brain and Behavior Research Foundation (to A.M.), by the Fondation pour la Recherche Médicale (Équipe FRM DEQ2013326488 to P.F.), by the French National Cancer Institute Grant TABAC-16–022 (to P.F.) by the Fédération de la Recherche pour le Cerveau (FRC Rotary Espoir en tête 2012 to P.F.) and by the Labex Bio-Psy. A.M. was recipient of a fundamental research prize from the Medisite Foundation for Neuroscience. R.D.C. was supported by a Ph.D. fellowship from the DIM Cerveau and Pensée program of the Région Ile-de-France, and by a fourth-year Ph.D. fellowship from FRM (FDT20170437427). D.L. was recipient of a Labex Biopsy post-doctoral fellowship. RHK was supported by grants from the NIH (U01-NS090527 and R01-NS100911). P.F. and A.M. laboratory is part of the École des Neurosciences de Paris Ile-de-France RTRA network. P.F. is member of LabEx Bio-Psy and of DHU Pepsy.

## Additional information

### Funding

| Funder | Grant reference number | Author |
|---|---|---|
| DIM Cerveau Pensée | PhD fellowship | Romain Durand-de Cuttoli |
| Fondation pour la Recherche Médicale | PhD fellowship FDT20170437427 | Romain Durand-de Cuttoli |
| Labex | Bio-Psy post-doctoral fellowship | Damien Lemoine |
| National Institutes of Health | U01-NS090527 | Richard H Kramer |

| Fondation pour la Recherche Médicale | Equipe FRM DEQ2013326488 | Philippe Faure |
| Institut National Du Cancer | TABAC-16-022 | Philippe Faure |
| Fédération pour la Recherche sur le Cerveau | Rotary Espoir en tête 2012 | Philippe Faure |
| Brain and Behavior Research Foundation | NARSAD Young Investigator Grant | Alexandre Mourot |
| Agence Nationale de la Recherche | ANR-JCJC | Alexandre Mourot |
| Fondation de France | Fondation Medisite fundamental research prize | Alexandre Mourot |
| National Institutes of Health | R01-NS100911 | Richard H Kramer |

The funders had no role in study design, data collection and interpretation, or the decision to submit the work for publication.

### Author contributions

Romain Durand-de Cuttoli, Data curation, Formal analysis, Methodology, Writing—original draft, Writing—review and editing, Performed viral injections and cannula implantations, Slice patch-clamp experiments, In vivo electrophysiology, Behavioral studies and immunochemistry; Sarah Mondoloni, Data curation, Formal analysis, Methodology, Writing—original draft, Writing—review and editing, Performed cell culture and in vitro electrophysiology, Slice patch-clamp experiments and immunochemistry; Fabio Marti, Data curation, Formal analysis, Performed in vivo electrophysiology; Damien Lemoine, Data curation, Formal analysis, Performed cell culture and in vitro electrophysiology; Claire Nguyen, Data curation, Performed immunochemistry; Jérémie Naudé, Formal analysis; Thibaut d'Izarny-Gargas, Data curation, Performed slice patch-clamp experiments; Stéphanie Pons, Uwe Maskos, Dirk Trauner, Resources; Richard H Kramer, Resources, Funding acquisition, Writing—review and editing; Philippe Faure, Conceptualization, Supervision, Funding acquisition, Project administration, Writing—review and editing; Alexandre Mourot, Conceptualization, Data curation, Supervision, Funding acquisition, Methodology, Writing—original draft, Project administration, Writing—review and editing

### Author ORCIDs

Romain Durand-de Cuttoli  https://orcid.org/0000-0003-0240-7608
Jérémie Naudé  https://orcid.org/0000-0001-5781-6498
Thibaut d'Izarny-Gargas  https://orcid.org/0000-0002-6084-5836
Stéphanie Pons  https://orcid.org/0000-0003-1027-0621
Philippe Faure  https://orcid.org/0000-0003-3573-4971
Alexandre Mourot  http://orcid.org/0000-0002-8839-7481

### Ethics

Animal experimentation: All experiments were performed in accordance with the recommendations for animal experiments issued by the European Commission directives 219/1990, 220/1990 and 2010/63, and approved by Sorbonne Université.

### Decision letter and Author response

Decision letter https://doi.org/10.7554/eLife.37487.020
Author response https://doi.org/10.7554/eLife.37487.021

## Additional files

### Supplementary files

• Transparent reporting form
DOI: https://doi.org/10.7554/eLife.37487.018

## Data availability

All data generated or analyzed during this study are included in the manuscript and supporting files. Source data are provided for Figures 1 to 5.

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
