## [Decision Letter]

Thank you for submitting your article "Somatic nicotinic acetylcholine receptors control the activity of dopamine neurons and reward-related behaviors" for consideration by *eLife*. Your article has been reviewed by three peer reviewers, and the evaluation has been overseen by a Reviewing Editor and Gary Westbrook as the Senior Editor. The following individuals involved in review of your submission have agreed to reveal their identity: Robert Wickham (Reviewer #2) and Stephanie J Cragg (Reviewer #3). The reviewers have discussed the reviews with one another and the Reviewing Editor has drafted this decision to help you prepare a revised submission.

Summary of reviews and post-review discussion

Durand de-Cotelli et al., uses a highly innovative approach to assess the β2*nAChR's responsiveness to nicotine in vivo as well as to assess this receptor's function in nicotine reward. Previous knockout work is limited due to potential compensatory changes by loss of the β2*nAChR, and pharmacological treatment non-selectively targets non DAergic neurons as well as DAergic neurons, making it challenging to assign a direct function to DAergic β2*nAChRs. This approach may be able to get around these two problems by directly activating/inactivate somatic β2*nAChRs in the VTA. In general, the reviewers were appreciative of the study, but they all agreed that the manuscript suffers from key overinterpretations which limited their enthusiasm. In a nutshell, they questioned the claim (repeatedly made throughout the manuscript and in the title) that the manipulation is specific to "somatic" β2*nAChRs". The original reviews are summarized below with the essential points that must be addressed in the revision:

1) It is not clear why the actions described here couldn't involve other non-DA cells and afferents that input to the recorded VTA DA neurons. There is no cell-specific targeting. From the description provided in the manuscript, it seems as if the virus containing the β2E61C construct is transfected in all neurons in the VTA, which should mean pre and post-synaptic neurons should be activated upon light stimulation. Thus, it is unclear how the manipulation only target "somatic", i.e. postsynaptic, β2*nAChRs. Of note, similar lentiviral methods which the authors cite (Maskos et al., 2005) result in expression in at least DA and GABA neurons in VTA. The authors Figure 4—figure supplement 1A also shows strong GFP where there is no TH (e.g. bottom right). Even the β2^-/-^ manipulation is not targeted to DA neurons but is global. The authors strongly intimate postsynaptic "somatic" nAChRs on DA neurons but other sources have simply not been excluded or controlled for. Thus, the authors must provide experimental evidence that their β2E61C is localized to "somatic" compartments. In the CPP experiments notably, direct demonstration of an absence of presynaptic β2*nAChR contribution is necessary (a possible control could be to show that eGFP is not expressed in pedunculopontine and laterodorsal tegmental nuclei).

2) What is the evidence that stereotaxic injection of PGK-β2E61C-Ires-EGFP lentivirus in the VTA does not transduce presynaptic cholinergic afferents or GABAergic neurons of the VTA? The authors argue that the control of DA firing is only on the control of the somatic post synaptic β2*nAChR expressed in DA VTA neurons but it lacks the demonstration that their stereotactic injection of PGK-β2E61C-Ires-EGFP lentivirus in the VTA does not transduce presynaptic cholinergic afferents. It could be easily done by showing that eGFP is not expressed in pedunculopontine and laterodorsal tegmental nuclei.

3) In the Discussion, the authors revisit an old debate to distinguish the role of nAChRs in VTA versus DA terminals in nicotine reinforcement because "Nicotine can also induce DA release through pre-synaptic effects, either directly in the striatum at the level of DA terminals". They are revisiting an old argument to which they have previously contributed and which many have previously addressed. It has previously been discussed that the actions of nicotine at axon terminals appear to involve primarily desensitization (not inducing release) which magnifies the effects of a change in DA neuron firing rate (previous works by Dani, Maskos, Faure, Exley, Cragg). Terminal effects of nAChRs might depend upon changes in DA neuron firing rate. The authors have again revisited the mistakes of the past and concluded that receptors outside of VTA are not important for nicotine reinforcement, which they have not shown here. The writing needs to be more accurate.

---

## [Author Response]

Summary of reviews and post-review discussion

*Durand de-Cotelli et al., uses a highly innovative approach to assess the β2*nAChR's responsiveness to nicotine* in vivo *as well as to assess this receptor's function in nicotine reward. Previous knockout work is limited due to potential compensatory changes by loss of the β2*nAChR, and pharmacological treatment non-selectively targets non DAergic neurons as well as DAergic neurons, making it challenging to assign a direct function to DAergic β2*nAChRs. This approach may be able to get around these two problems by directly activating/inactivate somatic β2*nAChRs in the VTA. In general, the reviewers were appreciative of the study, but they all agreed that the manuscript suffers from key overinterpretations which limited their enthusiasm. In a nutshell, they questioned the claim (repeatedly made throughout the manuscript and in the title) that the manipulation is specific to "somatic" β2*nAChRs". The original reviews are summarized below with the essential points that must be addressed in the revision.*

We’d like to thank the editors and reviewers for their positive feedback on our technology. We agree that our viral strategy does not allow specific targeting of VTA DA neurons, and that photo-manipulation is not restricted to “somatic” receptors. Hence, we removed the word “somatic” from the title. In fact, we changed the title to add emphasis to the technology. The new title is: “Manipulating midbrain dopamine neurons and reward-related behaviors with light-controllable nicotinic acetylcholine receptors”. In addition, we now clearly show that LinAChR expression is restricted to VTA neurons and is absent in afferents from other brain areas. Please see below for a point-by-point answer to the reviewers’ concerns. We combined the answers to points 1 and 2, since they were tightly related.

1) It is not clear why the actions described here couldn't involve other non-DA cells and afferents that input to the recorded VTA DA neurons. There is no cell-specific targeting. From the description provided in the manuscript, it seems as if the virus containing the β2E61C construct is transfected in all neurons in the VTA, which should mean pre and post-synaptic neurons should be activated upon light stimulation. Thus, it is unclear how the manipulation only target "somatic", i.e. postsynaptic, β2*nAChRs. Of note, similar lentiviral methods which the authors cite (Maskos et al., 2005) result in expression in at least DA and GABA neurons in VTA. The authors Figure 4—figure supplement 1A also shows strong GFP where there is no TH (e.g. bottom right). Even the β2^-/-^ manipulation is not targeted to DA neurons but is global. The authors strongly intimate postsynaptic "somatic" nAChRs on DA neurons but other sources have simply not been excluded or controlled for. Thus, the authors must provide experimental evidence that their β2E61C is localized to "somatic" compartments. In the CPP experiments notably, direct demonstration of an absence of presynaptic β2*nAChR contribution is necessary (a possible control could be to show that eGFP is not expressed in pedunculopontine and laterodorsal tegmental nuclei).2) What is the evidence that stereotaxic injection of PGK-β2E61C-Ires-EGFP lentivirus in the VTA does not transduce presynaptic cholinergic afferents or GABAergic neurons of the VTA? The authors argue that the control of DA firing is only on the control of the somatic post synaptic β2*nAChR expressed in DA VTA neurons but it lacks the demonstration that their stereotactic injection of PGK-β2E61C-Ires-EGFP lentivirus in the VTA does not transduce presynaptic cholinergic afferents. It could be easily done by showing that eGFP is not expressed in pedunculopontine and laterodorsal tegmental nuclei.

The reviewers are right: there is no specific cell targeting, and we apology if the first version of the paper made it sound like there was. We used a lentiviral vector with a ubiquitous pGK promoter to express β2 LinAChR in VTA neurons (both DA and non-DA cells). Non-DA VTA neurons include GABAergic interneurons; hence, LinAChRs are potentially expressed both at somato-dendritic and at terminal (i.e. pre-synaptic) sites within the VTA. Importantly, we have verified that they were absent in afferents that input VTA neurons. We modified the manuscript in several ways to make this clear. First, we changed Figure 2B (immunocytochemistry) and the associated text to clearly show that, within the VTA, GFP could be found both in tyrosine hydroxylase (TH) positive and negative (i.e. DA and non-DA) neurons. We also added a scheme in Figure 2 to illustrate the expression profile of LinAChRs within the VTA. Second, we modified the Introduction slightly to clearly explain where native nAChRs are expressed (within the VTA and on extra-VTA afferents) and added one reference (Grady et al., 2007); the original sentence lacked precision. Third, we show that GFP is absent in LDT and PPN afferent neurons (new Figure 2—figure supplement 1), in agreement with the absence of retrograde transport capacities of lentiviral vectors (Mazarakis et al., 2001). Hence LinAChRs are only expressed in intra-VTA neurons. Finally, since photocontrol is specific to neurons of the VTA but could involve pre-synaptic nAChRs on GABAergic interneurons, we changed the title (as mentioned above) and modified the Results and Discussion sections where necessary, to avoid using the words “somatic” or “post-synaptic” as much as possible. We hope this new version of the manuscript addresses the reviewer’s concerns satisfactorily.

3) In the Discussion, the authors revisit an old debate to distinguish the role of nAChRs in VTA versus DA terminals in nicotine reinforcement because "Nicotine can also induce DA release through pre-synaptic effects, either directly in the striatum at the level of DA terminals". They are revisiting an old argument to which they have previously contributed and which many have previously addressed. It has previously been discussed that the actions of nicotine at axon terminals appear to involve primarily desensitization (not inducing release) which magnifies the effects of a change in DA neuron firing rate (previous works by Dani, Maskos, Faure, Exley, Cragg). Terminal effects of nAChRs might depend upon changes in DA neuron firing rate. The authors have again revisited the mistakes of the past and concluded that receptors outside of VTA are not important for nicotine reinforcement, which they have not shown here. The writing needs to be more accurate.

We agree with the reviewer that nAChRs outside the VTA, and especially those expressed in the striatum, are very important for nicotine reinforcement. We did not aim at revisiting an old debate, rather we aimed at illustrating the fact that our technology makes it possible to manipulate nAChRs in specific sub-cellular compartments, for instance intra-VTA neurons, without affecting nAChRs in DA terminals. This is clearly not possible with other techniques such as viral re-expression strategies, and we wanted to emphasize this aspect. Reviewer is right concerning our lack of precision though. We modified the paragraph in the Discussion according to the reviewer’s comments. Notably we modified the sentence concerning the effects of nicotine at the level of DA terminals, which now is: “nicotine also modulates DA release by desensitizing nAChRs expressed in the striatum at the level of DA terminals”. We also changed “main” by “key” players concerning β2 nAChRs of the VTA. We hope reviewer will be satisfied by the new version.